**www.cambridge.org/ext**

megafauna; paleoclimate; anthropogenic impacts; Quaternary extinctions; subsistence shift hypothesis

**Corresponding author:**
Laurie R. Godfrey;
Email: lgodfrey@umass.edu

# Patterns of late Holocene and historical extinctions on Madagascar

Laurie R. Godfrey[1] , Zachary S. Klukkert[2] , Brooke E. Crowley[3,4] ,
Robin R. Dawson[5] , Peterson Faina[6,7] , Benjamin Z. Freed[8] ,
Evon Hekkala[9,10] , Cortni Borgerson[11] , Harimanjaka A. M. Rasolonjatovo[12] ,
Patricia C. Wright[13] and Stephen J. Burns[5]

[1]Department of Anthropology, University of Massachusetts Amherst, Amherst, MA, USA; [2]Department of Anatomy and Cell Biology, Oklahoma State University Center for Health Sciences, Tulsa, OK, USA; [3]Department of Geosciences, University of Cincinnati, Cincinnati, OH, USA; [4]Department of Anthropology, University of Cincinnati, Cincinnati, OH, USA; [5]Department of Earth, Geographic and Climate Sciences, University of Massachusetts Amherst, Amherst, MA, USA; [6]Columbia Climate School, Columbia University in the City of New York, New York, NY, USA; [7]Olo Be Taloha Lab, Lamont-Doherty Earth Observatory, Palisades, New York, NY, USA; [8]Department of Language & Cultural Studies, Anthropology, and Sociology, Eastern Kentucky University, Richmond, Kentucky, USA; [9]Department of Biological Sciences, Fordham University, Bronx, New York, NY, USA; [10]American Museum of Natural History, New York, New York, NY, USA; [11]Department of Anthropology, Montclair State University, Montclair, NJ, USA; [12]Mention Bassins sédimentaires, Evolution, Conservation, Université d'Antananarivo, Faculté des Sciences, Antananarivo, Madagascar and [13]Department of Anthropology, Stony Brook University, Stony Brook, New York, NY, USA

## Abstract

Around 1000 years ago, Madagascar experienced the collapse of populations of large vertebrates that ultimately resulted in many species going extinct. The factors that led to this collapse appear to have differed regionally, but in some ways, key processes were similar across the island. This review evaluates four hypotheses that have been proposed to explain the loss of large vertebrates on Madagascar: Overkill, aridification, synergy, and subsistence shift. We explore regional differences in the paths to extinction and the significance of a prolonged extinction window across the island. The data suggest that people who arrived early and depended on hunting, fishing, and foraging had little effect on Madagascar's large endemic vertebrates. Megafaunal decline was triggered initially by aridification in the driest bioclimatic zone, and by the arrival of farmers and herders in the wetter bioclimatic zones. Ultimately, it was the expansion of agropastoralism across both wet and dry regions that drove large endemic vertebrates to extinction everywhere.

## Impact statement

This review addresses how changes in climate and human activities have impacted the habitats and vertebrate communities of Madagascar over the past several thousand years. The great "Red Island" is currently experiencing a biodiversity crisis that threatens the survival of numerous endemic vertebrate species including virtually all of its lemurs. The island has already lost its largest-bodied terrestrial vertebrates. We show that climate, specifically extended drought, severely affected only the driest part of Madagascar. Early hunters and foragers had minimal impact, but a shift to agropastoralism had major consequences across Madagascar. It was not the arrival of foragers but changes in human livelihoods that truly threatened this island's biodiversity one thousand years ago and continue to do so today. Our review documents the deep roots of today's accelerating biodiversity crisis, helping us to better understand current risks to endemic species and challenges to the resilience of people with diverse livelihoods.

## Introduction

During the late Holocene, Madagascar experienced a suite of vertebrate extinctions encompassing species belonging to seven orders of mammals, eight orders of birds, and two orders of reptiles (Supplementary Table 1). The timing of population collapse and extinction is not well documented for all species, but many are known to have survived into the past two millennia, some into the past millennium, and at least seven likely survived European contact ~500 years ago. Humans clearly threaten global biodiversity today, and they have been considered the primary drivers of large-vertebrate extinctions over the past 125,000 years (Smith et al., 2018; Andermann et al., 2020). Over this coarse scale, extinction rates are correlated with the time of human colonization globally; faunal diversity loss and extinction rates lack correlation with climate change but are well predicted by human population density (Smith et al., 2018; Andermann et al.,

2020). Landmasses that were colonized early by humans (such as Australia and the Americas) experienced early megafaunal loss. Islands were, with some exceptions, among the last places to be colonized by humans, and those likely colonized during the Holocene (including Madagascar) experienced postcolonization rapid loss of large-bodied vertebrates (Louys et al., 2021; Tomlinson et al., 2024).

However, when viewed on a finer scale, a more complex picture of Holocene extinction on islands emerges. New evidence (presented in this article) that Madagascar's megafaunal collapse occurred millennia after initial human colonization has forced researchers to reconsider the threats posed by colonizing hunters/foragers. There are debates over whether Madagascar's earliest human colonizers underwent population crashes of their own, the relative impacts of different human subsistence strategies (e.g., foraging vs. farming, or hunting and/or fishing vs. herding), the importance of climate change, and the existence of regional variation in the trajectories of megafaunal decline. Megafaunal extinctions on Madagascar could resemble those on islands colonized by humans during the Pleistocene (e.g., Flores, Timor), which show little evidence of temporal association with initial human arrival (Louys et al., 2021).

Our review of late Holocene extinctions in Madagascar begins by describing four extinction hypotheses that propose different primary triggers: Overkill; aridification; synergy; and subsistence shift. We next review research on the chronology of: (1) megafaunal population collapse and extinction; (2) initial human arrival; and (3) the arrival and spread of agropastoralism. Next, we describe modern Madagascar's climate and vegetation diversity, and temporal changes in climate and habitat. We then evaluate how well the available data support each hypothesis. Finally, we note how ethnohistoric records can improve our understanding of extinction processes, and how a better understanding of Madagascar's extinctions may help secure a future for this island's remaining vertebrate biodiversity.

## Hypotheses concerning large-vertebrate extinction in Madagascar

### Overkill hypothesis

The overkill or hunting hypothesis (Martin, 1967, 1984) maintains that rapid extinction followed colonization by hunter/foragers of major bodies of land, including Madagascar (Walker, 1967). Hunter/foragers quickly over-exploited large-bodied, endemic species that were naïve to human predators.

### Aridification hypothesis

The aridification hypothesis, in contrast, holds that drought is the primary driver of megafaunal extinction in Madagascar. Mahé and Sourdat (1972) questioned the importance of hunting in Madagascar, arguing instead that this island's late Holocene extinctions began under the influence of drought prior to human arrival. Decades later, a second version of the aridification hypothesis postulated that vertebrate populations declined well *after* humans arrived – coinciding instead with a presumed island-wide drought that peaked ~950 years ago (Virah-Sawmy et al., 2010).

### Synergy hypothesis

Burney (1997) proposed the synergy hypothesis as an alternative to overkill and aridification. In its earliest, simplest form, this hypothesis suggests that the late Holocene extinctions were triggered by a complex combination of events involving both climate change and human activities. A more detailed version of this hypothesis was formulated by Burney et al. (2003, 2004). It posits a particular sequence of events: (1) drought prior to human arrival made megaherbivores (hippopotamuses, elephant birds, tortoises) vulnerable to human impacts; (2) megaherbivore population collapse occurred during the first half of the 1st millennium CE (where CE = Common Era, i.e., the past 2000 years) at the hands of human hunters; (3) natural fires spiked during the ensuing centuries, triggered by the growth of fire-prone vegetation that would have been regulated by now-decimated megaherbivores; and (4) large-bodied domesticated animals such as cattle and sheep were later introduced by agropastoralists. Two key tenets of the synergy hypothesis are that habitat change followed (and did not precede) the loss of endemic, large-bodied herbivores, and that livestock were introduced after the megafaunal collapse.

### Subsistence shift hypothesis

Finally, the subsistence shift hypothesis (Godfrey et al., 2019; Hixon et al., 2021a; Godfrey and Douglass, 2022), building on an earlier "biological invasion" hypothesis (Dewar, 1984, 1997), posited a direct connection between megafaunal population collapse and the arrival and subsequent spread of agropastoralism across Madagascar, associated with the expansion of the Indian Ocean trade network. It suggests that the foragers who had colonized Madagascar prior to the introduction of agropastoralism had little impact on the megafauna because their populations were small. The subsistence shift from hunting/foraging to herding/farming triggered rapid habitat modification due to forest clearance and deliberately set fires. Perhaps counterintuitively, hunting of endemic wild animals increased with the spread of livestock husbandry, not merely because agropastoralism supported rapid human population growth but because herders have less motivation than hunters and fishers to avoid overexploitation of animals on which they no longer depend (wild meats only supplement foods acquired mainly through farming and herding).

## The chronology of megafaunal population decline, collapse, and extinction in Madagascar

The timing of the megafaunal collapse in Madagascar is widely understood to have occurred within the past 2000 years (Crowley, 2010), but there is disagreement over precisely when, within that span, the megafauna declined, populations collapsed, and species went extinct. Regional decline for some species has been reconstructed at 2000 years or earlier. Some studies place megafaunal collapse shortly after 2000 years ago and extinction almost 1000 years later, while others place megafaunal collapse ~1000 years ago and extinction later still.

Burney et al. (2003, 2004) reconstructed the timing of megafaunal collapse using counts in sediments of spores of *Sporormiella* (a coprophilous fungus that they accepted as a proxy for megafaunal abundance; see also Raper and Bush, 2009). They estimated the megafaunal collapse to have occurred relatively early in the 1st millennium CE. Last occurrence dates for fossils belonging to different species were used to infer megafaunal extinction near the end of the 1st millennium CE.

Event estimation techniques, constructing confidence intervals for extinction events based on distributions of available dates, have become preferred tools of analysis. Standard statistical tools for

inferring confidence intervals have been modified to take into consideration dating uncertainty and variation in the number of available dates (Bradshaw et al., 2012). Using these tools, Hixon et al. (2021a) estimated an extinction window from 1200 to 700 cal yr BP (where cal yr BP = calibrated year Before Present, and "present" = calendar year 1950) for giant lemurs, elephant birds, hippos, and giant tortoises. Modifying these tools further to estimate introduction windows for domesticated animals, these authors discovered remarkable concordance between the extinction of endemic vertebrates and the introduction of domesticated vertebrates. Both occurred during the same 500-year-long period, in association with an expanded Indian Ocean trade network.

Godfrey et al. (2019) and Faina et al. (2021) used odds ratio analysis to capture the trajectory of megafaunal loss and estimate the timing of megafaunal collapse. This technique employs maximum likelihood chi-square analysis with different temporal cutpoints to find the point at which the representation of extinct vs. extant taxa at subfossil sites changes the most. Precipitous changes in the ratio of extinct to extant species are taken as evidence of megafaunal population collapse. Because dated subfossil specimens belonging to extant as well as extinct species are included in this type of analysis, sample sizes are usually much larger than those based on extinct taxa only. Godfrey et al. (2019) found evidence of megafaunal collapse between ~1250 and 1050 cal yr BP across Madagascar. Faina et al. (2021) found that large-vertebrate populations in southwestern Madagascar collapsed slightly earlier than in northwestern Madagascar (~1100 vs. ~1000 cal yr BP, respectively). Extinction followed later in both regions.

## When did people first colonize Madagascar?

The timing of human colonization of Madagascar is debated, with estimates varying from over 10,000 years ago to slightly over 1000 years ago. Hansford et al. (2018, 2020) posited human arrival before 10,000 years ago on the basis of cutmarked bones, while other authors (Anderson et al., 2018; Mitchell, 2019) defended arrival 9000 years later. Genetic data (Pierron et al., 2017; Alva et al., 2022) suggest that today's Malagasy population descended primarily from people who arrived independently from Africa (~1500 years ago) and southeast Asia (over ~2000 years ago). Pierron et al. (2017) could not rule out the earlier arrival of people who did not contribute (or contributed trivially) to the modern gene pool, but they uncovered no direct evidence of such populations.

Evidence for the arrival of hunter/foragers prior to 4000 years ago comes from archaeologists, paleoecologists, and paleontologists, however. Stone projectiles and blades have been found at two rock shelters in the extreme north of Madagascar, Lakaton'i Anja and Ambohiposa (Dewar et al., 2013). Using optically stimulated luminescence (OSL), Dewar et al. (2013) found the basal cultural layers at Lakaton'i Anja to be over 4000 years old. Stone tools and remains of transported prey (terrestrial vertebrates from nearby forests as well as fish and shellfish from the coast several kilometers away) reveal intermittent use of the shelter by foragers. Bones from *Palaeopropithecus maximus*, a giant extinct lemur species recovered from a hearth at Lakaton'i Anja, range in age from 1545 to 1305 cal yr BP (Douglass et al., 2019). Inconsistencies between OSL dates for stone flakes and radiocarbon dates for charcoal in some sedimentary layers have been taken by critics to suggest that the dates assigned to tools in the basal layers are faulty (Anderson et al., 2018; Mitchell, 2019). It is likely, however, that the problem rests

instead with disturbance and movement of the organic matter from which radiocarbon dates were derived, given that the OSL-dated stone flakes are in correct stratigraphic order while the radiocarbon dates are not (Dewar et al., 2013; Wright, 2022).

Early archaeological sites also exist in the Velondriake Marine Protected Area in the coastal southwest (Douglass, 2016; Douglass et al., 2018). Excavations of Velondriake sites have unearthed fisher/forager economies with bony remains of fish, worked shells, and early ceramics. Almost all radiocarbon dates are from the first half of the 1st millennium CE (Douglass et al., 2018). There is also evidence of occupation by foragers before 3000 years ago (Davis et al., 2020). These are sites lacking ceramics that have not yet been excavated or directly dated but that were identified using satellite-based remote sensing. Davis et al. (2020) believe them to be older than 3000 years because they are at locations that would have been flooded between 3000 and 1000 years ago, but available to coastal fishers before 3000 years ago. If they were younger than 1000 years, they would have had ceramics, as do settlement sites younger than 1000 years elsewhere in the southwest (Wright, 2022).

Bones of extinct vertebrates with apparent butchery marks (cuts and chops at nonrandom locations on skeletal elements) have been found at sites in both wetter and drier parts of Madagascar (MacPhee and Burney, 1991; Perez et al., 2005; Gommery et al., 2011; Hansford et al., 2018, 2020; Godfrey et al., 2019; Hixon et al., 2021a, 2022; Vasey and Godfrey, 2022). These have been interpreted as evidence of meat processing, dismemberment, and bone marrow extraction. The temporal range of these modified bones (from >10,000 cal yr BP to ~1000 cal yr BP) is tremendous. However, modified bones are rare and lack spatial clustering until ~1200 cal yr BP. There is evidence for early, pre-agricultural translocation of wild animals, including some birds and bushpigs (Godfrey et al., 2019; Balboa et al., 2024).

Some scholars have dismissed all presumed butchery traces on bones that are older than ~1200 cal yr BP, claiming that they are either nonanthropogenic, or, if anthropogenic, then not ancient, but rather products of sloppy excavation damage (Anderson et al., 2018; Mitchell, 2019). However, a careful, recent excavation at a Velondriake site called Tampolove yielded hippo femora bearing chops very similar to those on disputed bones, and new dates for disputed bones (~3500 cal yr BP and ~1600 cal yr BP) have confirmed their antiquity (Hixon et al., 2022). This recent work provides support for the antiquity of other hippo bones with chops previously attributed to excavation damage. Apparently, hunter/foragers did indeed exploit megaherbivores in southwest Madagascar for thousands of years prior to the introduction of agropastoralism (Hixon et al., 2022).

## When did agropastoralists arrive on Madagascar and when did agropastoralism spread?

Small hamlets and villages with ceramics, iron tools, and middens revealing dependence on agropastoralism first appeared in northern Madagascar ~1300 years ago, which indeed corresponds to the timing of the expansion of the Indian Ocean trade network (Vérin and Chanudet, 1996). Crops like rice first cultivated in southeast Asia during the 1st millennium CE were introduced to the Comoros islands and soon thereafter to northern Madagascar (Kull et al., 2011; Crowther et al., 2016; Beaujard, 2017). Simultaneously, domesticates (e.g., cattle, sheep, goats, dogs, cats) arrived, mainly from Africa (Blench, 2007; Sauther et al., 2020; Hixon et al., 2021a,

2021b). By ~1000 years ago, large and small settlement sites (e.g., Mahilaka on the northwestern coast and Andranosoa in the extreme south) began proliferating across Madagascar (Wright and Fanony, 1992; Radimilahy, 1998; Parker Pearson, 2010; Radimilahy and Crossland, 2015; Godfrey and Douglass, 2022; Wright, 2022). As agropastoralism spread, contact grew between the previously small human populations that came to Madagascar from southern Africa and Southeast Asia, and their genetic admixture triggered sustained human population expansion (Alva et al., 2022). Communities also responded to local environmental crises by moving to places with better resources and reconfiguring or expanding social networks in order to gain access to coveted distant resources (Davis et al., 2023).

## Madagascar's climate and vegetation diversity

Madagascar has four major bioclimatic zones that differ in vegetation and hydroclimate (Figure 1). An eastern escarpment blocks most of the moisture carried by southeasterly trade winds and thus partly controls mean annual precipitation (MAP). Precipitation west of this divide derives primarily from the summer monsoon, which is controlled by seasonal changes in insolation, and the position of the tropical rain belt. This creates a latitudinal gradient with rainfall greater in the north than in the south. Rainfall is also greater in the highlands just west of the escarpment than it is toward the western coast. MAP is generally above 1000 mm/year in the three wetter bioclimatic zones ("humid", "subhumid", and "dry") (US Geological Survey, 2020); indeed, during the wet austral summer (December through February), mean daily rainfall in this region exceeds 8 mm and, in places, exceeds 12 mm, which places this region within the top 1% of global rainfall intensities (Jury, 2022). MAP tends to be considerably lower in the fourth ("subarid") bioclimatic zone, particularly in its southernmost portion, where it rarely surpasses 500 mm/year (USGS, 2020), and where mean daily rainfall during the austral summer can fall below 2 mm (Jury, 2022). Finally, the latitudinal moisture gradient characteristic of the island continues through the subarid zone, which is wetter in its more northern parts.

All bioclimatic zones today have degraded forests and unforested landscapes. Grassy biomes have expanded in the recent past, particularly in the three wetter zones, but bioclimatic zones also preserve relict forests (Figure 1, Vegetation Zones). In the humid bioclimatic

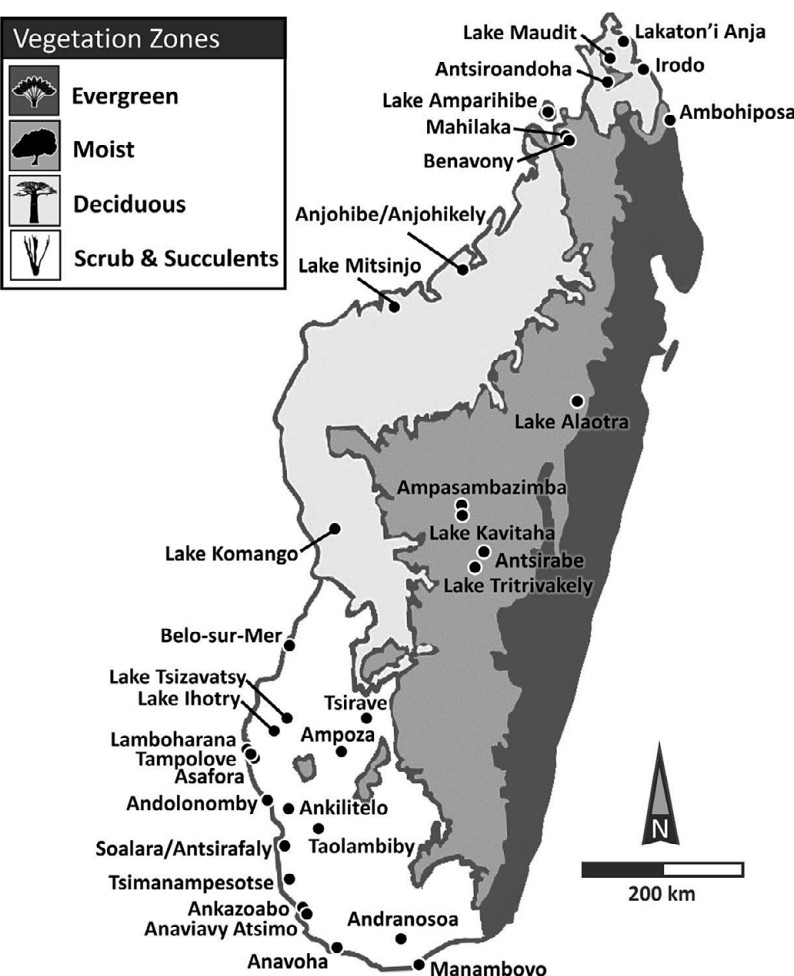

**Figure 1.** Locations of sites used to infer temporal changes in habitat and fauna and their likely triggers (climate or humans). Shown also are Madagascar's primary vegetation (or bioclimatic) zones, from wettest (darkest shading) to driest (lightest shading): (1) evergreen forest (humid bioclimatic zone) represented here by the traveler's palm (*Ravenala madagascariensis*); (2) moist forests (subhumid bioclimatic zone) represented here by the Tapia tree (*Uapaca bojeri*); (3) deciduous forest (dry bioclimatic zone) represented here by the baobab (*Adansonia digitata*); and (4) scrubland or spiny thicket and succulent woodland (subarid bioclimatic zone) represented by the Madagascar ocotillo (*Alluaudia procera*). "Wetter" regions include evergreen, moist, and dry deciduous forests, while "drier" regions include scrubland and succulent woodland. See text for details regarding differences in mean annual rainfall and seasonality. Vegetation/Bioclimatic Zones are modified from: http://www.efloras.org/web_page.aspx?flora_id=12&page_id=1204 (Accessed 12/11/2023)

zone, these forests are typically evergreen. The vegetation of the subhumid bioclimatic zone is diverse but generally moist; it includes closed-canopy evergreen and semi-evergreen forests as well as scler-ophyllous woodland. The dry bioclimatic zone in the northwest is dominated by deciduous forests. Finally, the subarid bioclimatic zone exhibits succulent woodland in its northern and eastern portions and spiny thicket or scrubland in its southern and southwestern portions (Burgess et al., 2004).

## The chronology of regional changes in climate and habitat

Figure 2 summarizes previous studies of late Holocene variation in hydroclimate and floral communities in the wet and dry regions of Madagascar and provides our own inferences regarding the pri-mary drivers of that variation (climate vs. humans), sometimes differing from those of the studies' authors.

The Holocene paleoclimate history of the wetter region of Madagascar is well documented due to an abundance of stalagmites (or speleothems) from Anjohibe cave in the northwest that grew during the Holocene, and well-studied lake cores in northern and central Madagascar (Figure 2, Supplementary Figures 1 and 2, Supplementary Table 2). Stalagmites can be analyzed for chrono-logical changes in stable oxygen and stable carbon isotope values ($\delta^{18}O$ and $\delta^{13}C$, proxies for rainfall and vegetation cover, respect-ively). Drought conditions are indicated when these proxies reveal a significant decline in rainfall coupled with a loss of woody $C_3$ plants and a concomitant increase in $C_4$ grasses, $C_4$ sedges, or succulent CAM plants. Stalagmites at Anjohibe (e.g., Burns et al., 2016; Scrox-ton et al., 2017; Dawson et al., 2024) reveal a relatively wet early Holocene and relatively dry middle and late Holocene. There was a dry interval ~9200 years ago followed by wetter periods ~8200 years ago attributed to a global event. Another shift from wetter to drier conditions began ~6000 and 5500 years ago at Anjohibe and Lake Maudit. Episodes of unusually dry hydroclimate characterized the mid-Holocene between 5000 and 3000 years ago. Wetter conditions prevailed again beginning 1600 years ago and persisted through the end of the 1st millennium CE (Supplementary Table 2).

New stalagmite records for the drier (subarid) region of Mada-gascar from Asafora, Mitoho, and Anaviavy Atsimo caves have greatly improved our understanding of the climate of Madagascar's southwest during the late Pleistocene and Holocene (Scroxton et al., 2019; Faina et al., 2021, 2024; Godfrey et al., 2021; Burns et al., 2022; Figure 2; Supplementary Figures 1 and 2, Supplementary Table 3). They reveal limited Holocene stalagmite growth until ~3600 BP, suggesting that the early and middle Holocene in the subarid zone may have been extremely dry. Stalagmites generally require at least 300 mm/year of rainfall to grow (Bar-Matthews et al., 1997). Beginning ~3600 years ago at Asafora, rainfall supported stalagmite growth, which means only that MAP likely exceeded 300 mm, which is still very dry. Oxygen isotopes ($\delta^{18}O$) reveal gradual drying following these relatively "wet" conditions. Begin-ning at 1700 BP, rainfall declined precipitously, resulting in true drought conditions at 1600 BP that ended ~900 BP. Oxygen isotopes for Anaviavy Atsimo, which is 260 km south of Asafora, indicate that this site was consistently drier than Asafora throughout the late Holocene, due to a strong latitudinal effect (Faina et al., 2024).

## Putting extinction hypotheses to the test

Below we evaluate the degree to which the human, climate, environment, and extinction histories in the wetter and drier regions support the four alternative extinction hypotheses (Figure 2, Supplementary Figures 1 and 2, and Supplementary Tables 2 and 3).

## The wetter bioclimatic zones

The overkill hypothesis is not supported in the wetter regions of Madagascar (Supplementary Table 2). There is clear evidence for the presence of hunter/foragers in northern Madagascar (Lakaton'i Anja) prior to 4000 years ago, but there is no evidence that these people triggered the megafaunal population collapse. In the wetter bioclimatic zones, megafaunal populations crashed precipitously ~1000 years ago, as estimated using odds ratio analysis (Figure 2). Using event estimation tools, Hansford et al. (2021) suggested that the largest of the megafauna (the megaherbivores, including hippo-potamuses and elephant birds) disappeared from the deciduous forests of the northwest between 2364 and 2078 cal yr BP. This inference is based on dates for nine hippos and one elephant bird from Anjohibe. While this is 1000 years prior to the broader regional megafaunal collapse, it is still too late to be explained by overkill. The trigger for this early apparent megaherbivore decline is uncertain. Hansford et al. (2021) attributed it to deforestation, but it is unclear why megaherbivores would be threatened by deforest-ation if they consumed monocots as well as browse (Crowley et al., 2021). There may be a better explanation, involving climate.

On the whole, the aridification hypothesis fares poorly as an explanation for megafaunal collapse in the wetter bioclimatic zones (Figure 2). It certainly does not explain the regional megafaunal collapse at the end of the 1st millennium CE. The hydroclimate of this region was relatively wet during the entire second half of the 1st millennium CE, and precipitous habitat change in this region occurred ~1100 years ago in the absence of climate change (Figure 2). However, aridification may help to explain the early local extirpation of hippos. In comparison to the early Holocene, the mid- and late Holocene at Anjohibe were relatively dry (Supplementary Table 2). This region of Anjohibe supported hip-pos until ~2000 years ago. It must have had streams or other water bodies that were perennial, as hippos require perennial water sources to meet their thermoregulatory needs (Jablonski, 2004). There are no perennial bodies of water in the immediate vicinity of Anjohibe today. It is impossible to know when exactly streams near Anjohibe became seasonal, but one can ask whether the apparent local extirpation of hippopotamuses occurred during a period of episodic drought. New climate records at Anjohibe cover the millennium prior to local hippopotamus decline and more (Williams et al., 2024; Dawson et al., 2024; Figure 2). Dry events did indeed occur in the region of Anjohibe prior to the local disappearance of hippos (Figure 2). There is no reason to believe that all streams and lakes in the wetter region became seasonal at that time, as large perennial lakes and streams persist there today. But drought may have resulted in hippo extirpation at locations with *smaller*, more vulnerable lakes or streams, without impacting access to perennial water elsewhere.

Like the aridification hypothesis, support for the synergy hypothesis in the wetter region is also mixed (Figure 2, Supplementary Table 2). Only a few sites in this region meet the expectations of this hypothesis. As stated above, the synergy hypothesis predicts that Madagascar experienced the decimation of megaherbivore populations by hunters (evidenced by declines in the spores of *Sporormiella*) followed by increases in natural fire (evidenced by spikes in charcoal particles). Both should occur after colonization by hunter/foragers but well prior to the introduction of agropastoralism ≤1300 years ago. Whereas charcoal spikes have

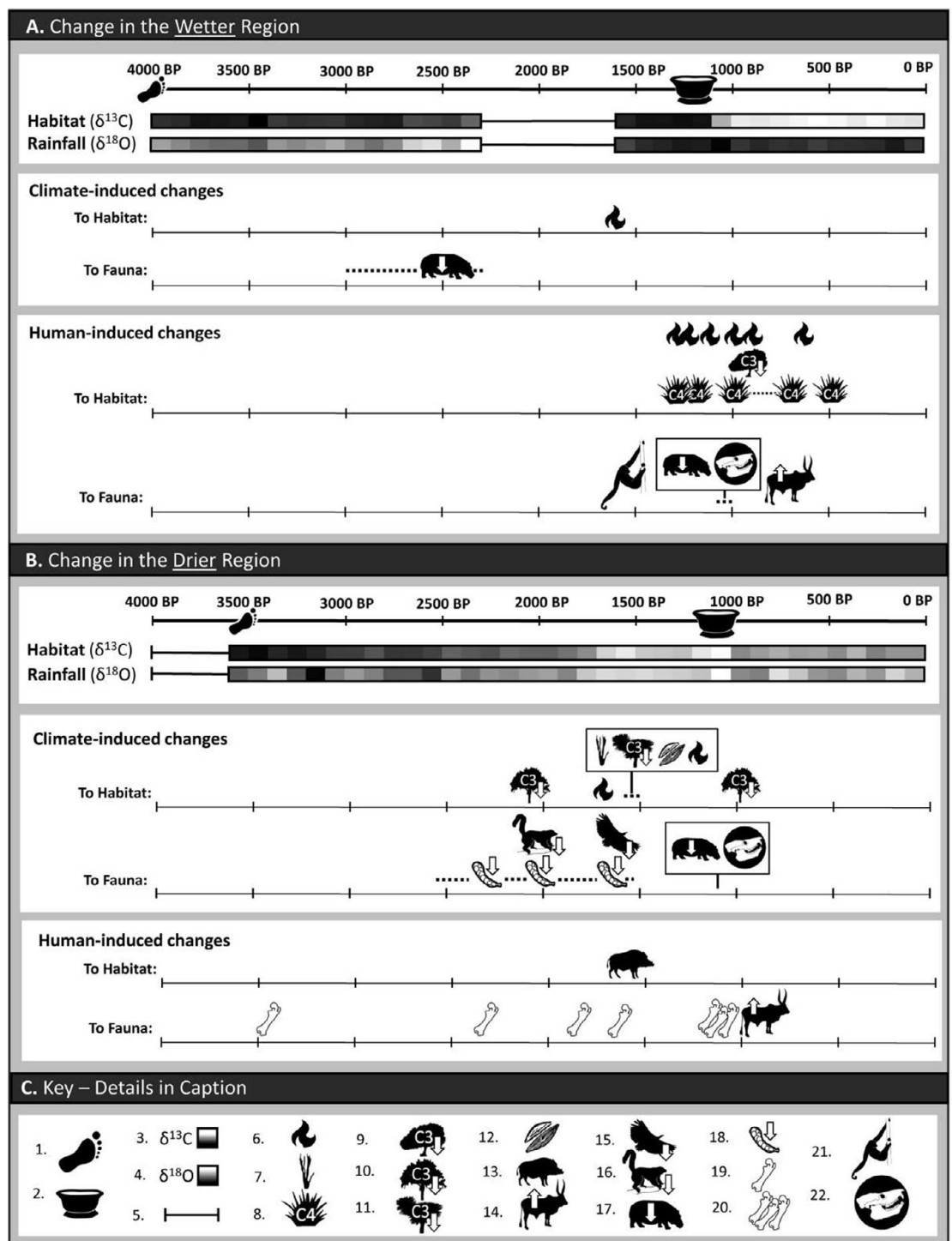

**Figure 2.** Climate- and human-induced changes in habitat and fauna in the wetter and drier regions of Madagascar over the past 4000 years, inferred from a variety of records (see text, Supplementary Figures 1 and 2, and Supplementary Tables 2 and 3 for notes and published sources). For each region, high-resolution stable isotope records from single cave sites (Anjohibe in the wetter region and Asafora in the drier region) are provided; these are $\delta^{13}C$ (a proxy for ground vegetation) and $\delta^{18}O$ (a proxy for rainfall amount). Icon key: 1) Foragers (hunters/fishers/gatherers) present; 2) Sites with ceramics appear, followed by evidence of introduced plants and domesticated animals, small hamlets, large ports, large settlements and urbanization; 3) High-resolution $\delta^{13}C$ records (from stalagmites AB2, AB11, and AB13 at Anjohibe in the wetter region and stalagmite AF2 at Asafora in the drier region) displaying evidence of changes in habitat (darker shading reflects more $C_3$ plants and lighter shading reflects more $C_4$ and/or CAM plants); 4) High-resolution $\delta^{18}O$ records (from stalagmites AB2, AB11, and AB13 at Anjohibe in the wetter region and stalagmite AF2 at Asafora in the drier region) displaying evidence of changes in rainfall (with darker shading signaling more rainfall and lighter shading signaling less); 5) High-resolution data covering the designated time interval do not exist; 6) Direct evidence of fire from charcoal particles in sediments; 7) Palynological evidence of increase in $C_4$ and/or CAM plants; 8) Palynological evidence of increase in $C_4$ grasses; 9) Palynological evidence of decrease in $C_3$ trees; 10) Palynological evidence of decrease in palms; 11) Palynological evidence of decrease in *Pandanus* (screw pines); 12) Changes in diatom conductivity suggest lake salinization; 13) Malagasy bushpig (*Potamochoerus larvatus*) genome suggests introduction from Africa by 1500 BP; 14) Presence of cattle inferred from dated *Bos* bones and/or increase in *Sporormiella* spores; 15) Decline in freshwater birds (e.g., *Icthyophaga vociferoides*, the Madagascar fish eagle) inferred from subfossil record; 16) Decline in large-bodied arboreal lemurs inferred from subfossil record; 17) Decline in megaherbivores inferred from subfossil record; 18) Decline in faunal community biomass inferred from decline in *Sporormiella* spores; 19) Butchery traces on isolated bones of extinct vertebrate species; 20) Butchery traces on bones of extinct vertebrate species, observed in temporal concentration at sites such as Tsirave; 21) Bones of the extinct lemur, *Palaeopropithecus*, in cultural context; 22) Regional collapse of endemic large-vertebrate populations (represented by the skull of giant extinct lemur, *Megaladapis*).

been recorded in lake sediments at many sites in the wetter bioclimatic zones (Supplementary Table 2), few (Lake Komango, and earlier at Lake Alaotra; see Broothaerts et al., 2023) occurred prior to the arrival of agropastoralism. Burney et al. (2003, 2004) interpreted the Komango charcoal spike as support for the synergy hypothesis. There is no immediately prior *Sporormiella* decline at Komango or any other site in the wetter bioclimatic zones. However, Lake Komango is in the southwestern part of the wetter bioclimatic zones (Figure 1), and is only ~190 km north of Belo-sur-Mer, a subarid site that experienced early *Sporormiella* decline. The charcoal spike at Lake Komango can be interpreted as supporting the synergy hypothesis. Alternatively, as we will discuss below in considering habitat change in the drier region of Madagascar, both the charcoal spike at Lake Komango and the decline in *Sporormiella* spores at Belo-sur-Mer could reflect broad regional drying (more pronounced to the south) at ~1600 years ago.

Further north, at Anjohibe, there is additional possible support for hunters impacting habitat prior to the introduction of agropastoralism. This is based on the notion that a lack of correlation (i.e., a "decoupling") of stalagmite $\delta^{13}C$ and $\delta^{18}O$ values could indicate that changes in vegetation are not driven by changes in rainfall. Wang et al. (2019) reported a decoupling of stalagmite $\delta^{18}O$ and $\delta^{13}C$ values between 2500 and 1500 yr BP at Anjohibe, which they interpreted as support for human impact. However, the overall fluctuation in values for both isotopes during this time is ≤1.5‰ and the net change in $\delta^{18}O$ is ~0.03‰. These values likely reflect local stasis in both rainfall and vegetation. A decoupling of $\delta^{18}O$ and $\delta^{13}C$ values at Anjohibe that occurred centuries later (between 1100 and 1000 yr BP), after the introduction of agropastoralism, was far more dramatic. It involved an increase in $\delta^{13}C$ by 10.5‰ in ~100 years, without concomitant change in $\delta^{18}O$ values. At that time, $C_3$ woody plants plummeted and $C_4$ grasses rose sharply while rainfall showed trivial net change (Figure 2; Supplementary Figures 1 and 2, Supplementary Table 2). This decoupling provides strong support for human impact associated with the arrival of agropastoralism, not with foraging. It supports the subsistence shift hypothesis.

There is additional strong support for the subsistence shift hypothesis in the wetter bioclimatic zones. Multiple proxies all converge to suggest that agropastoralism triggered habitat change and a wholesale shift in vertebrate fauna beginning in the late 1st and extending into the 2nd millennium CE. Carbon isotope data from vertebrate fossils recovered from Anjohibe, palynological and charcoal records from regional lake cores, and spikes in *Sporormiella* spores at Lakes Amparihibe and Kavitaha, all bear witness to environmental changes associated with agropastoralism (Figure 2, Supplementary Table 2). They correlate quite well with the expansion of the Indian Ocean trade network, the introduction of cultivars and domesticated animals, and the megafaunal population collapse ~1000 years ago. They also fall comfortably within the megafaunal extinction window (1200 to 700 cal yr BP) constructed by Hixon et al. (2021a) using event estimation techniques.

### The subarid (driest) bioclimatic zone

The overkill hypothesis is not supported in the dry region of Madagascar (Figure 2, Supplementary Table 3). There is strong evidence of the presence of hunters, foragers, and fishers here for thousands of years, yet odds ratio analysis, in broad agreement with extinction event estimation, suggests that megafaunal populations did not crash until ~1100 years ago. Therefore, the overkill hypothesis can be rejected for this bioclimatic zone as it can for the wetter zones.

Multiple lines of evidence support the aridification hypothesis in the subarid bioclimatic zone (Figure 2, Supplementary Table 3). As this region became progressively drier, prior to the spread of agropastoralism, we find: (1) *Sporormiella* spore and/or paleontological evidence for vertebrate species decline at Belo-sur-Mer, Tampolove, Andolonomby, Taolambiby, Tsimanampesotse, and Soalara/Antsirafaly; (2) palynological evidence for a loss of $C_3$ trees and increased dominance of $C_4$/CAM plants at Lake Ihotry, Andolonomby, and Taolambiby; (3) evidence for salinization at Lakes Ihotry and Tsimanampesotse; (4) evidence for increased fire at Taolambiby, Belo-sur-Mer, and Andolonomby; (5) evidence at Andolonomby for megafauna foraging in drier habitats, based on nitrogen isotope values; and (6) coupled changes in $\delta^{18}O$ and $\delta^{13}C$ values from speleothems at Asafora and Anaviavy Atsimo Caves, suggesting changes in habitat driven by climate (Supplementary Table 3).

The synergy hypothesis appears to be supported at two sites where declines in *Sporormiella* spores are followed by spikes in charcoal, all during the first half of the 1st millennium CE (and thus well before the arrival of agropastoralism). These are Belo-sur-Mer (1907.5 cal yr BP, then ~1700 years ago) and Andolonomby (1607.5 cal yr BP, then ~1500 years ago) (Supplementary Table 3). However, as noted above, the aridification hypothesis likely provides a better explanation for early megafaunal decline in the subarid bioclimatic zone when the expectation of the synergy hypothesis (that habitats changed only after the *Sporormiella* decline) is not met. The dramatic changes in habitat at Lake Ihotry (lake salinization), Andolonomby (sharp decline in *Pandanus* and other $C_3$ trees, increase in $C_4$ sedges), and Taolambiby (decline in $C_3$ trees and increase in $C_4$ and CAM plants) were coeval with the precipitous drought recorded at Asafora ~1600 years ago (Figure 2, Supplementary Table 3).

While there is strong support for the aridification hypothesis in the subarid bioclimatic zone, recent paleoclimate research suggests that droughts similar or worse in magnitude and duration to that of the 1st millennium CE were hardly unusual for this part of Madagascar (Faina et al., 2024). Given that the very species that were driven to extinction in the late Holocene did survive previous droughts, it is hard to make the argument that climate alone was responsible for their extinction. Droughts may have triggered declines in large-vertebrate populations during the 1st millennium CE and earlier, but it was likely agropastoralism that drove these taxa to extinction, mostly during the 2nd millennium CE, as populations of farmers and herders in the subarid bioclimatic zone increased. Bodies of fresh water, scarce or seasonal because of the drought, would have become magnets not merely for endemic herbivores and their natural predators, but for people with domesticated animals including hunting dogs (Clarke et al., 2006; Godfrey et al., 2019; Hixon et al., 2021a, b). Effectively, agropastoralists changed the landscape for endemic vertebrates. Between 1200 and 1000 cal yr BP, we find sites near water bodies with high concentrations of butchered megafaunal bones (Vasey and Godfrey, 2022). These were likely wild-meat processing localities or trapping locations for targeted taxa, such as the now-extinct lemur, *Pachylemur insignis*. By ~1000 years ago, the relative abundance of cutmarked bones of wild extant species increased at such sites (Sullivan et al., 2022), and by ~800 cal yr BP, their faunal composition is entirely extant vertebrates (Figure 2, Supplementary Table 3).

### Late survivors provide evidence for a protracted extinction window

Ethnohistoric data (apparent eye-witness accounts and lore recorded in Madagascar during the past 500 years) have typically

been ignored as evidence of the timing of megafaunal extinction in Madagascar, but these data exist, and they document a prolonged extinction window. The last occurrence dates confirm 2nd millennium survival for nine now-extinct species (Table 1). These include seven lemurs (*Pachylemur insignis*, *Archaeolemur majori*, *A. edwardsi*, *Palaeopropithecus ingens*, *Megaladapis madagascariensis*, *M. grandidieri*, and *Daubentonia robusta*), hippos (*Hippopotamus* spp.), and elephant birds (likely *Aepyornis maximus*; Figures 3 and 4). There are recent eye-witness accounts for: *Pachylemur* in the late-20th century (Vasey and Godfrey, 2022; Borgerson, unpubl. data); *Archaeolemur* in the mid-20th century (Burney and Ramilisonina, 1998 on kidoky; Vasey et al., 2012); and hippopotamuses in the 19th and 20th centuries (Kaudern, 1915; Grandidier, 1971; Godfrey, 1986; Burney and Ramilisonina, 1998; Wright, 2016). Credible eye-witness accounts for *Palaeopropithecus* and elephant birds (likely *Aepyornis*) are slightly older, dating to the 17th century (de Flacourt, 1658; Hébert, 1998; Godfrey and Jungers, 2003). There are also apparent eye-witness reports (late 19th to 21st centuries) of two species that have no radiocarbon dates in the 2nd millennium CE: the giant fosa, *Cryptoprocta spelea* (Freed, 1996; Nomenjanahary et al., 2021) and the horned crocodile, *Voay robustus* (Vaillant and Grandidier, 1910; Sibree, 1915). One reported *Voay robustus* (skin, skull, and skeleton of a full adult) from Lake Alaotra, shipped to the natural history museum in Paris in the late 1800s, was catalogued and then described by Vaillant and Grandidier (1910). The lack of recent data for *C. spelea* and

*V. robustus* can be attributed to their limited dated sample sizes (n = 3 and 4, respectively).

Figures 3 and 4 show a striking lack of temporal and geographic correspondence between locations for the last occurrence of radiocarbon and ethnohistoric records. For example, ethnohistoric accounts place both *Pachylemur* and *Palaeopropithecus* most recently in the wettest bioclimatic zone, while radiocarbon dates place them most recently in the driest bioclimatic zone (Table 1). We interpret this discrepancy as a taphonomic artifact, with conditions for collagen preservation in bone far better in dry places, rather than a testimony to the often-presumed unreliability of ethnohistoric records, although both can apply. Of the ~500 calibrated dates for extinct vertebrates, ~80% are from the drier region, while over half of ethnohistoric record locations are in the wetter region.

## Conclusions: looking ahead

Of the four main hypotheses regarding megafaunal extinction, the subsistence shift hypothesis (which posits that the shift from hunting and foraging to herding and farming was the primary trigger for large-vertebrate population collapse) is best supported, particularly for the wetter region of Madagascar where there is strong evidence of rapid anthropogenic habitat modification in the absence of climate change. It is also clear that agropastoral activities affected the large vertebrates even in the driest parts of Madagascar, making survival in small refuges increasingly impossible.

**Table 1.** Regional "last occurrence" radiocarbon dates within the period of megafaunal decline for those extinct taxa that apparently survived into the last half of the 2nd millennium CE. Radiocarbon dates older than 2000 cal yr BP are excluded. Radiocarbon site locations (by number) are shown in Figure 4.

| Genus | Vegetation type | Site | Conventional age ($^{14}$C yr BP) | Calibrated age[1] (mean ± 1σ cal yr BP) | Location in Figure 4 | Source |
|---|---|---|---|---|---|---|
| *Cryptoprocta*[2] | Scrubland | Ankazoabo Cave | 1865 ± 35 | 1767.5 ± 67.5 | 1 | Crowley (2010) |
| *Pachylemur* | Moist forest | Ampasambazimba | 1300 ± 30 | 1172.5 ± 97.5 | 2 | Crowley (2010) |
| | Succulent woodland | Tsirave | 940 ± 20 | 820 ± 85 | 3 | Godfrey et al. (n.d.) |
| | Scrubland | Taolambiby | 1234 ± 27 | 1115 ± 60 | 4 | Anderson et al. (2018) |
| *Archaeolemur* | Dry deciduous forest | Antsiroandoha | 1020 ± 50 | 872.5 ± 87.5 | 5 | Simons et al. (1995) |
| | Succulent woodland | Tsirave | 1305 ± 35 | 1172.5 ± 97.5 | 6 | Crowley (2010) |
| | Scrubland | Soalara | 1090 ± 60 | 932.5 ± 132.5 | 7 | Godfrey et al. (2021) |
| *Palaeopropithecus* | Dry deciduous forest | Lakaton'i Anja | 1480 ± 30 | 1332.5 ± 42.5 | 8 | Douglass et al. (2019) |
| | Succulent woodland | Belo-sur-Mer | 2008 ± 90 | 1915 ± 210 | 9 | Burney et al. (2004) |
| | Scrubland | Ankilitelo | 510 ± 80 | 442.5 ± 122.5 | 10 | Simons (1997) |
| *Hippopotamus* | Moist forest | Antsirabe | 1215 ± 25 | 1077.5 ± 97.5 | 11 | Crowley (2010) |
| | Succulent woodland | Ampoza | 1325 ± 24 | 1227 ± 127.5 | 12 | Hansford et al. (2021) |
| | Scrubland | Lamboharana | 1100 ± 15 | 942.5 ± 17.5 | 13 | Hixon et al. (2021a) |
| *Aepyornis* | Dry deciduous forest | Irodo | 1150 ± 90 | 1080 ± 180 | 14 | Mahé and Sourdat (1972) |
| | Moist forest | Antsirabe | 1349 ± 28 | 1232 ± 52.5 | 15 | Hansford et al. (2021) |
| | Scrubland | Manambovo | 840 ± 80 | 767.5 ± 137.5 | 16 | Battistini et al. (1963) |
| *Mullerornis* | Succulent woodland | Belo-sur-Mer | 1280 ± 60 | 1167.5 ± 117.5 | 17 | Burney (1999) |
| | Scrubland | Antsirafaly | 1270 ± 25 | 1118 ± 53 | 18 | Faina et al. (2021) |
| *Voay* | Succulent woodland | Ampoza | 1460 ± 30 | 1322.5 ± 42.5 | 19 | Hekkala et al. (2021) |

[1]Conventional $^{14}$C dates were calibrated or recalibrated to calendar years before present (cal yr BP) using Calib 8.2 (Stuiver and Reimer, 1993) and the southern hemisphere calibration curve (SHCal20; Hogg et al., 2020).
[2]The genus *Cryptoprocta* has an extant representative (*C. ferox*). This entry refers specifically to the extinct *C. spelea*.

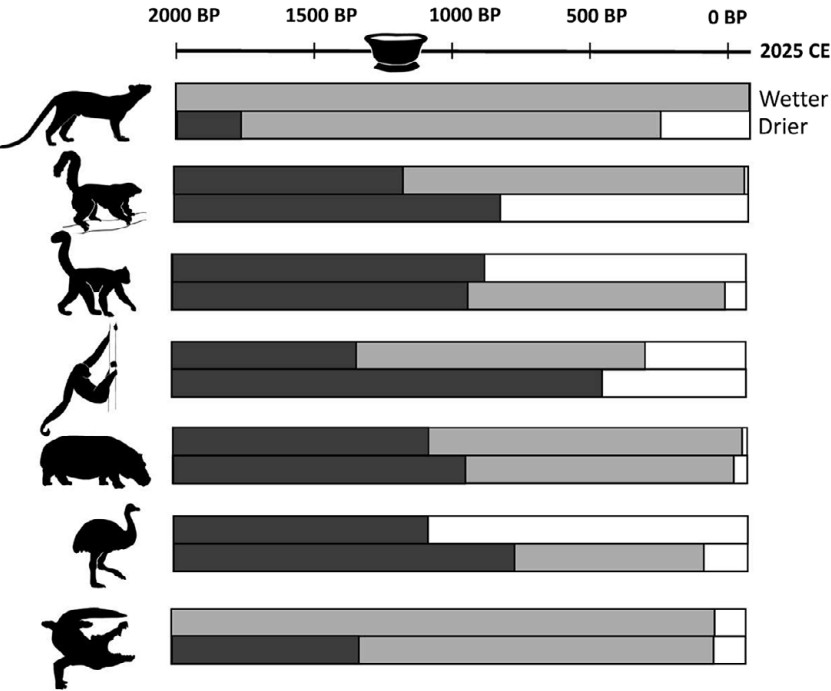

**Figure 3.** Comparison of (1) black bars: subfossil radiocarbon dates (terminating at most recent records) and (2) gray bars: ethnohistoric records (terminating at most recent credible eye-witness reports) for seven extinct vertebrate taxa in both wetter and drier regions of Madagascar. White bars represent no data. Taxa are, from top to bottom: *Cryptoprocta spelea*, *Pachylemur* spp., *Archaeolemur* spp., *Palaeopropithecus* spp., *Hippopotamus* spp., elephant birds (*Aepyornis* spp. or *Mullerornis modestus*), and *Voay robustus.*

The aridification hypothesis (that drought triggered megafaunal collapse) is partly supported. There is strong evidence that late Holocene aridification in the driest part of Madagascar negatively impacted megafaunal populations prior to the arrival of agropastoralism, but there was no island-wide drought at the time of the megafaunal collapse.

The synergy hypothesis in its basic form (which suggested that late Holocene extinctions resulted from a combination of natural and human impacts) is supported, but support is weak for the particular sequence of events posited by the more detailed model (i.e., decimation of megaherbivores via hunting prior to charcoal particle spikes and ecological transformation).

Finally, the overkill hypothesis (rapid megafaunal population collapse due to unsustainable hunting immediately following human colonization) is not supported. However, unsustainable hunting cannot be dismissed as contributing to megafaunal collapse, albeit counterintuitively at the hands of agropastoralists rather than hunter/foragers. The earliest people to arrive in Madagascar (likely over 4000 years ago) were hunters, foragers, and fishers with a stone tool culture. While they hunted large-bodied endemic vertebrates, their impact on megafaunal populations was small. Had it not been for the later introduction of agropastoralism – a subsistence strategy that promoted habitat modification (clearing, burning) and triggered increased hunting pressure – population recovery likely would have been possible for many endemic large-vertebrate species. The narrative that human arrival inevitably triggers rapid faunal collapse on islands fails for Madagascar, as it fails for islands colonized by humans during the Pleistocene (Louys et al., 2021).

We draw two important inferences from our data review. First, genetic bottlenecks likely occurred regularly in the past without triggering extinction. Secondly, a severely reduced probability of population recovery during the 2nd millennium CE likely explains why so many species did go extinct in the very late Holocene, and why the number of Endangered or Critically Endangered extant species is increasing today at a remarkable rate, not merely in Madagascar, but across the globe (e.g., Verry et al., 2023).

Genetic research has revealed both recent and remote bottlenecks in extant species. Those more remote in time were likely driven by natural climate change (e.g., Parga et al., 2012; Quéméré et al., 2012; Salmona et al., 2017), but human activities have also caused population declines and genetic bottlenecks (e.g., Olivieri et al., 2008; Craul et al., 2009; Ranaivoarisoa et al., 2010; Lawler, 2011; Salmona et al., 2017; Sullivan et al., 2022; Fordham et al., 2024), often at a pace that exceeds that of climate change (Morelli et al., 2020; Michielsen et al., 2023). Human activities now threaten 98% of Madagascar's lemurs with imminent extinction, making them the world's most threatened vertebrate taxa (IUCN, 2020). Thus, whereas the triggers that led to the population collapse and extinction of large-bodied vertebrates in Madagascar (e.g., habitat modification, hunting) continue to imperil smaller-bodied extant vertebrate species (Borgerson et al., 2022), new and unprecedented risks to these species' survival (e.g., increased hunting pressure, more rapid deforestation) have emerged with the expanding human population, resulting in greater reliance on forests (Borgerson et al., 2019). Furthermore, increasingly complex social networks have facilitated human access to distant resources.

Despite differences in the extinction triggers in wetter and drier bioclimatic zones of Madagascar, in some ways, the regional extinction stories are similar. In all bioclimatic zones, megafauna that survived in remote pockets well into the 2nd millennium CE ultimately went extinct, regardless of climate or the persistence of essential resources. The failure of an increasing number of vertebrate species to survive the 2nd millennium CE underscores the importance of understanding the effects of increasingly complex human subsistence strategies. Malagasy communities continue to depend

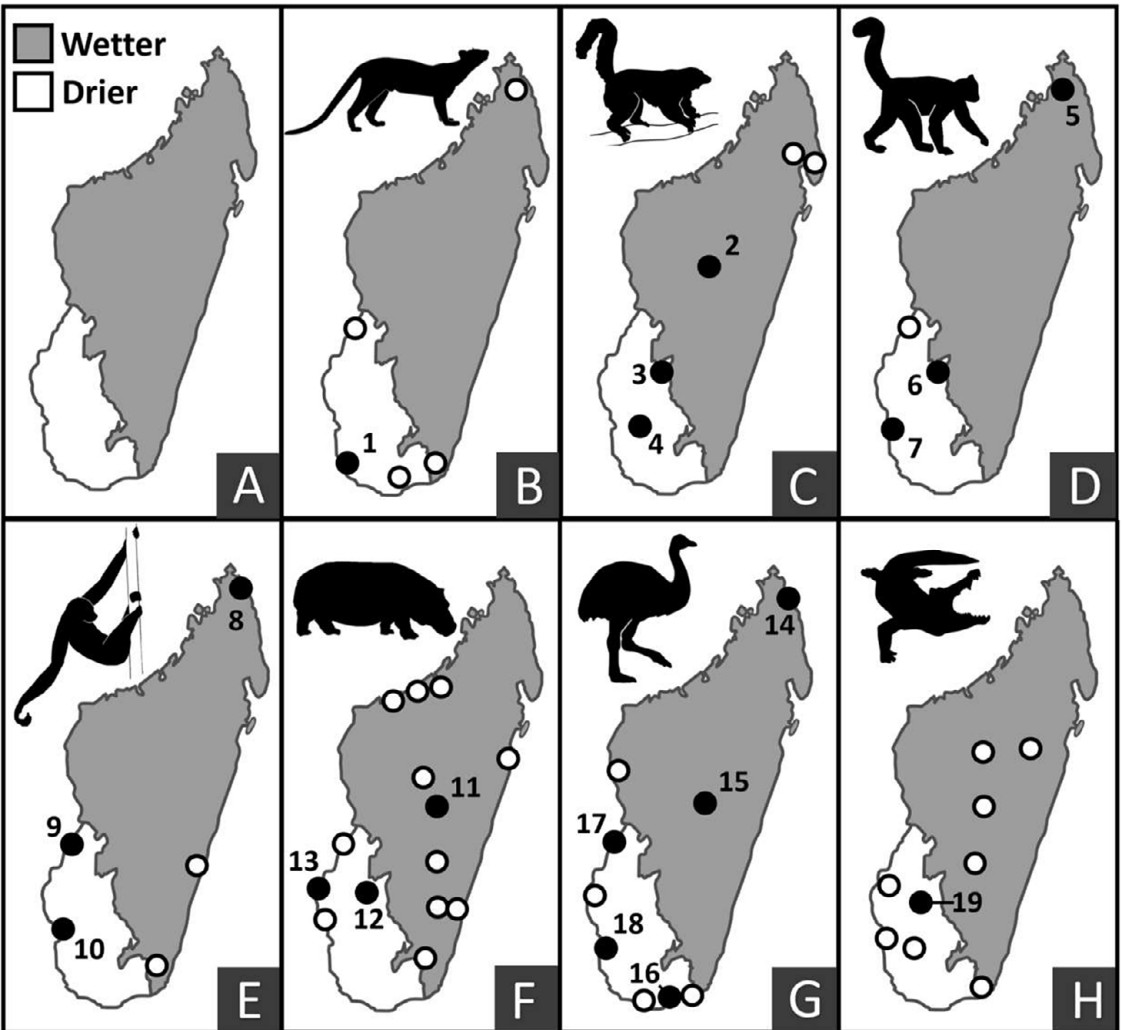

**Figure 4.** Comparison of the geographic locations of (1) last occurrence dates on subfossil bones (solid numbered circles) and ethnohistoric records (unfilled circles), the latter signaling persistence into the past 500 years (shown for large-bodied extinct vertebrates in both wetter and drier regions of Madagascar). Last occurrence records that fall outside the undisputed temporal range for possible megafaunal decline (the past 2000 years) are excluded. Panel A: Wetter and drier regions of Madagascar. Panel B: *Cryptoprocta spelea*; Panel C: *Pachylemur* spp.; Panel D: *Archaeolemur* spp.; Panel E: *Palaeopropithecus* spp.; Panel F: *Hippopotamus* spp.; Panel G: *Aepyornis* spp. or *Mullerornis modestus*; and Panel H: *Voay robustus.* Numbers on sites of last radiometric occurrence dates correspond to locations provided in Table 1.

on farming and herding today, but the growing number of extant species affected by human-induced genetic bottlenecks bears testimony to the urgency of improving the sustainability of human livelihoods, as they themselves change.

**Open peer review.** To view the open peer review materials for this article, please visit http://doi.org/10.1017/ext.2024.19.

**Supplementary material.** The supplementary material for this article can be found at http://doi.org/10.1017/ext.2024.19.

**Data availability statement.** No new data were collected for this review.

**Author contribution.** LRG conceptualized and designed this review. LRG and BEC wrote the manuscript, with contributions from ZSK, RRD, and CB. PF, RRD, and SJB provided input on stalagmite data; BEC, HR, and EH provided input on paleontological records; and CB, BZF, EH, and PCW provided input on ethnohistoric records. ZSK and RRD prepared the figures in the main text and supplement, with input from LRG, BEC, and SJB. LRG and BEC prepared the main text and supplementary tables. All authors reviewed, edited, and approved the manuscript.

**Financial support.** This review was funded by grants from the National Science Foundation (NSF BCS-1750598 to LRG and SJB; NSF AGS-2102923 and AGS-1702891 to SJB; NSF BCS-1749676 to BEC; NSF BCS-1749211 to Kathleen Muldoon who supported HAMR; and NSF DEB-1931213 to EH).

**Competing interest.** The authors declare no conflict of interest.

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
