## [Reviewer Report]

This study provides an overview and evaluation of different hypotheses to explain Holocene vertebrate extinctions in Madagascar. Drawing on the fossil and subfossil record, ethnohistoric data and paleoclimatic records from local sites across multiple bioclimatic zones on the island, the authors provide a nuanced view on the different hypotheses. In the end, the subsistence shift hypothesis (extinctions linked to agropastoralism) is favoured, but with a balanced discussion of the alternatives.

The study is interesting, thought-provoking and succeeds at bringing disciplines together. The text is well written and easy to follow.

I have a few suggestions for improvement:

- The impact statement ends quite abruptly without stating the results. Perhaps this is how it is meant to be according to the journal guidelines for this section, but if not, I suggest adding the key findings.

- L476 – not clear which species the adult individual belongs to.

- In the description of the subsistence shift hypothesis, the authors clearly explain how agropastoralism can counterintuitively lead to overhunting. Later in the manuscript, the overkill hypothesis is ruled out. Somehow this can be confusing, leading to the impression that overhunting was not important. I suggest adding in the concluding sections that overhunting is an integral part of the subsistence shift hypothesis.

- The figures could contain a bit more information in the way of synthesis. Figure 1 is useful for seeing where the different sites are located, but as a main text figure it does not provide much additional information for the reader (those working on Madagascar will have seen the bioclimatic zones map multiple times). Figure 2 provides an interesting perspective on the geographical distribution of fossil record sites versus ethnohistoric record sites. However, beyond the temporal distinction between these 2 types of records, somehow more information on the timing of the individual records could be provided, making it a more informative figure.

---

## [Reviewer Report]

The review by Godfrey and colleagues addresses a major long-standing question of Madagascar natural history, the demise of its megafauna. The manuscript is clearly written, well-organized and scientifically sound. Its topic, of high interest beyond the communities of biologist working in the south-western Indian Ocean, on island-biogeography and on extinction, is well fit to the targeted journal. I therefore recommend its publication, pending that the authors address the few major and minor points I raise below.

Major comments:

The authors do a great job in reviewing the extensive literature about the demise of its megafauna. They furthermore recall alternative hypotheses proposed in Madagascar and nicely evaluate the significance of accumulating evidences for each hypothesis. However, the manuscript remains visually poor, which may limit its impact. The large amount of diverse information allowing reconstructing Madagascar last millennia’s history would benefit from being visually synthesized, instead of being presented in tables. Similarly, the paper digestibility would rise if the tested hypotheses and their respective evidences were summarized visually. I therefore encourage the authors to consider such limited effort.

Minor comments:

Abstract: L58-59, “These same factors are present today and threaten the island’s remaining

biodiversity.” The factors mentioned here have not been defined yet. Moreover, the authors do compare alternative hypotheses it is unclear to me which same factors are present today. Climate changes have now been induced by human activities, hunter-gatherers were replaced by farmers and today’s farming is not driven by the same forces which led to the spread of agrospastoralism shift in Madagascar a millennium ago. The ‘global economy shift’ should not be confounded with the ‘spread of agropastoralism’. Although some activities driven by the global economy include agriculture and pastoralism practices, the drivers and their dynamic are different. It is therefore misleading to present current drivers of biodiversity loss as the ‘same triggers’ as those that lead to the demise of megafauna. The authors may want to refine their narrative about this, in particular in a specialized journal such as Extinction. Interestingly, it is again a shift in human economy, which is triggering the current wave of biodiversity loss. This idea, the authors vaguely mention at line 528, could be better presented in the discussion and resonate with the Neolithic transition in Madagascar.

From L111: The authors indistinctly use through the text the terminology “arrival of agropastoralism”, without distinguishing it from the ‘spread of agropastoralism’ for which they review proxies. Although it is not problematic in the introduction, it becomes little accurate in face of the reviewed literature (e.g. L379, L428) which is considered a proxy of its spread.

L204-209: The more recent paper from Alva et al., 2022 should also be used here too, to describe our knowledge of the chronology of Madagascar’s colonization by humans.

L281-294: some paragraphs are precisely rooted in the literature, while others (such as this one (L281-294) seems to ignore it, as if the information presented was common knowledge. In particular, the deforestation history / grassland expansion is highly relevant to the present review (and debated too) and should be precisely presented while being linked to the topic of the review. For instance, the grassy biomes expansion sentence cannot be left unreferenced.

L320-321: the term “holocene growth” is a bit out of context and therefore unclear in this sentence. Furthermore, it is unclear how limited “holocene growth” suggests that an area was dry. Please rephrase and clarify the causality.

L330-331: is the latitude responsible for Anaviavy Antsimo being drier, or a rainfall pattern somehow correlated with latitude?

L335 on: The bioclimatic organization of this whole section is confusing, while the header stipulate “Subhumid and dry”, the first sentence starts with a statement about ‘wetter regions of Madagascar’, followed by ‘wetter bioclimatic zones’ (L339), ‘dry bioclimatic zones’ (L345), ‘same bioclimatic zone’ (L347), ‘wetter bioclimatic zone’ (L353), ‘dry bioclimatic zone’ (L359). This makes the logical flow slightly hard to follow. I do not have the solution for the authors, but I am convinced they can find a way to present this in a more straightforward bioclimatic manner.

L481-482, 485: The authors may want to reformulate. The authors call “occurrence dates” (l481) or “dates” (L485) radiocarbon dated subfossil records, and compare it to ethnohistoric records. Besides being dated by technology, radiocarbon dated subfossil record are not superior to ethnohistoric ones. Both are dated and both have values and caveats. Both should therefore be named accurately (e.g. subfossil records and ethnohistoric records), without suggesting that certain are ‘dates’ and others are not.

L507, 528: As mentioned in the abstract, the ‘global economy shift’ should not be confounded with the ‘spread of agropastorlism’. Although some activities driven by the global economy include agriculture and pastoralism, the drivers and their dynamic are different. It is therefore misleading to present current drivers of biodiversity loss as the ‘same triggers’ as those that lead to the demise of megafauna. The authors may want to refine their narrative about this, in particular in a specialized journal such as Extinction. Interestingly, it is again a shift in human economy, which is triggering the current wave of biodiversity loss. This idea, the authors vaguely mention at line 528, could be better presented in the discussion and resonate with the Neolithic transition in Madagascar.

L510: to my understanding, Salmona et al., 2017 revisited the analyses of Quéméré et al., 2012 with better fitting models and providing a more nuanced view of the role of climate and humans.

Fig1: Considering the topic of the review, fig1 title’s should rather be “Map of locations of sites ….., showing bioclimatic zone’ (i.e. not the other way round)

Cited literature:

Alva, O., Leroy, A., Heiske, M., Pereda-Loth, V., Tisseyre, L., Boland, A., Deleuze, J.F., Rocha, J., Schlebusch, C., Fortes-Lima, C. and Stoneking, M., 2022. The loss of biodiversity in Madagascar is contemporaneous with major demographic events. Current Biology, 32(23), pp.4997-5007.

Salmona, J., Heller, R., Quéméré, E. and Chikhi, L., 2017. Climate change and human colonization triggered habitat loss and fragmentation in Madagascar. Molecular Ecology, 26(19), pp.5203-5222.

---

## [Editor Report]

Both reviewers found your paper of interest, and recommended publication with relatively minor corrections. Chief amongst those are the provision of a figure or figures that illustrate the volume of data you present in your tables. I agree that a figure would significantly add to the impact of your review. To their points, which I ask you address individually, I would add three of my own. First, while you do a great job shifting through different extinction hypotheses for Madagascar, you provide no critical introduction to megafauna extinctions globally in your introduction. I would suggest the science on this is far from settled: see for example Stewart, M., Carleton, W.C. and Groucutt, H.S., 2021. Climate change, not human population growth, correlates with Late Quaternary megafauna declines in North America. Nature Communications, 12(1), p.965, but several other recent publications in dfiferent parts of the world are equally critical of humans as the only cause of extinctions. At the risk of promoting my own research, I would also encourage the authors to look at Louys, J., Braje, T.J., Chang, C.H., Cosgrove, R., Fitzpatrick, S.M., Fujita, M., Hawkins, S., Ingicco, T., Kawamura, A., MacPhee, R.D. and McDowell, M.C., 2021. No evidence for widespread island extinctions after Pleistocene hominin arrival. Proceedings of the National Academy of Sciences, 118(20), p.e2023005118, which has relevance to your study, not least because it may provide some possibilities on how to move forward with your figure(s). Second, the statement on line 173 that the dung fungus is only associated with large mammals is incorrect. It is associated with biomass - the association with large mammals is only incidental, as large mammals are positively associated with large biomass. Smaller animals, in sufficient numbers, could also produce high concentrations of the fungus. Finally, a very minor point, but you need to provide a reference regarding hippos on lines 360-361.

---

## [Editor Report]

Thank you for addressing all reviewer and editor comments. I don’t think this needs to go back out for review, and can be accepted following these very minor edits. 

I am not sure how graphical your graphical abstract is - the impact is mainly in written as opposed to graphical form. I get it - this is a very tough subject to encapsulate in one figure, but I would be inclined to remove the current version if a simpler and less text-heavy version can’t be constructed.

Small factual error - lines 71-73 is incorrect. There is very early evidence of humans colonising islands, including modern humans colonising islands that have never been connected to any continents (e.g. Flores, Timor), with this occuring well before the colonisation of the Americas. Further, I would say that at a finer scale (line 75), island (as opposed to Holocene) extinctions become more complex. Conversely, Holocene extinctions on islands have traditionally been viewed as largely and simply (if not exclusively) caused by people (Louys and Tomlinson refs, contra your current wording). 

Line 208, and elsewhere - Wright reference - remove initial

Line 282, and elsewhere - your delta symbols have been lost 

Line 459 - Vaillant and G. Grandidier, remove initial